# Long-term health and human capital effects of in utero exposure to an industrial disaster: a spatial difference-in-differences analysis of the Bhopal gas tragedy

Gordon C McCord ,[1] Prashant Bharadwaj,[2] Lotus McDougal ,[3] Arushi Kaushik,[2] Anita Raj[3]

¹School of Global Policy and Strategy, University of California San Diego, La Jolla, California, USA
²Department of Economics, University of California San Diego, La Jolla, California, USA
³Center on Gender Equity and Health, Department of Medicine, University of California San Diego, La Jolla, California, USA

**Correspondence to**
Dr Gordon C McCord;
gmccord@ucsd.edu

## ABSTRACT

**Objectives** Globalisation and industrialisation can increase economic opportunity for low/middle-income nations, but these processes may also increase industrial accidents and harm workers. This paper examines the long-term, cohort-based health effects of the Bhopal gas disaster (BGD), one of the most serious industrial accidents in history.

**Design** This retrospective analysis uses geolocated data on health and education from India's National Family Health Survey-4 (NFHS-4) and the 1999 Indian Socio-Economic Survey by the National Sample Survey Organization (NSSO-1999) to examine the health effects of exposure to the BGD among men and women aged 15–49 years living in Madhya Pradesh in 2015–2016 (women n=40 786; men n=7031 (NFHS-4) and n=13 369 (NSSO-1999)), as well as their children (n=1260). A spatial difference-in-differences strategy estimated the relative effect of being in utero near Bhopal relative to other cohorts and to those further from Bhopal separately for each dataset.

**Results** We document long-term, intergenerational impacts of the BGD, showing that men who were in utero at the time were more likely to have a disability that affected their employment 15 years later, and had higher rates of cancer and lower educational attainment over 30 years later. Changes in the sex ratio among children born in 1985 suggest an effect of the BGD up to 100 km from the accident.

**Conclusions** These results indicate social costs stemming from the BGD that extend far beyond the mortality and morbidity experienced in the immediate aftermath. Quantifying these multigenerational impacts is important for policy consideration. Moreover, our results suggest that the BGD affected people across a substantially more widespread area than has previously been demonstrated.

## INTRODUCTION

Globalisation and industrialisation can increase economic opportunity for low/middle-income nations, but recent

---

### STRENGTHS AND LIMITATIONS OF THIS STUDY

⇒ This study uses different birth cohorts and a geographical difference-in-differences strategy to attribute causality of subsequent health and social consequences to the Bhopal gas disaster.

⇒ The design nets out any difference in outcomes that arise due to time-invariant aspects of being close to Bhopal, as well as aspects of belonging to a specific birth cohort.

⇒ As with any ecological exposure assignment, the cohort of in utero children we assign to exposure will include a range of actual exposure to methyl isocyanate gas.

⇒ Mortality can affect the composition of the population observed post-facto; if the weakest children in utero were more likely to die from the incident, then our estimates could be considered a lower bound.

⇒ The long-term consequences that we estimate could be the result of both direct effects from exposure as well as lack of subsequent mitigation of the effects through health, disability and education services.

---

cross-national evidence indicates that these processes may also be increasing industrial accidents and harming the workers of these nations.[1] Evidence from India suggests high numbers of industrial accidents, with the highest associated attributable deaths in the world.[2 3] Despite recommendations for improved protection for workers and industry in India,[3] observers (including United Nations experts) have stated that proposed rollbacks of environmental and worker protections violate the safety and health of workers and the environment, with potential impact on subsequent generations.[4–6] Unfortunately, little research is available on these multigenerational impacts, and inadequate recognition of these far-reaching

repercussions can diminish the memory of such tragedies for policy consideration. This research provides insight into these impacts on children born to female survivors of one of the worst industrial disasters in history, the Bhopal gas disaster (BGD).

The BGD occurred in December 1984 and involved a methyl isocyanate (MIC) gas leak at a Union Carbide pesticide plant near the city of Bhopal, in Madhya Pradesh, India. The leak spread toxic gas in approximately a 7 km radius around the plant, exposing more than half a million people and resulting in up to 30 000 deaths in the region.[7] There is a broad spectrum of serious long-term and chronic health consequences for hundreds of thousands of survivors, including children, manifesting across multiple systems (eg, respiratory, neurological, musculoskeletal, ophthalmic, endocrine).[7–10] These impacts may be the tip of the iceberg however, given that MIC toxins affected groundwater[11] and the reproductive health and other health outcomes of exposed women,[8] factors suggesting that generations not exposed to the toxic gas directly may nevertheless suffer adverse health and social impacts of the BGD event. Potential in utero effects cannot be understated. Studies document a fourfold increase in the rate of miscarriage following the gas leak, as well as increased risk of stillbirth and neonatal mortality.[9 10] Decades after the disaster, menstrual abnormalities and premature menopause have emerged as common problems among exposed women and their female offspring.[9] MIC has also been shown to damage human chromosomes.[12] Early clinical studies on the gas-affected population showed signs of increased chromosomal aberrations that could manifest as cancer,[13] though population-level changes in cancer rates 8 years after the accident were suggestive but not significant.[14]

This paper estimates the long-term health (specifically, adult cancer rates and disability) and human capital impacts (educational attainment) on individuals who were exposed to the BGD in utero or as children in 1984. Understanding the long-term impacts of industrial disasters on children, especially those in utero at the time of the disaster, is important for several key reasons. First, while the immediate damage to children in the form of direct health impacts is perhaps easier to assess, the long-term, health-related damages could take years to manifest and might not be part of the legal damages considered. Second, economic research over the last few decades has shown that interventions that improve outcomes for children have high rates of return[15]; to this second point, we include education as an important indicator of socioeconomic impact on these children, particularly as these secondary effects may compound health impacts. Our findings offer timely information to inform debate on potential rollbacks of policies related to social and environmental protections attached to industrialisation in India and elsewhere.

## METHODS

### Data

The principal data source is the fourth round of the Demographic and Health Surveys (DHS) in India, which was conducted between 2015 and 2016 (also called National Family Health Survey, or NFHS-4, in India).[16] The DHS/NFHS-4 is a high-quality household survey that provides detailed information on health at the individual level, and is representative at both the national and state levels. All interviews in Madhya Pradesh were completed in 2015. For the first analysis focusing on differences in the sex ratio, the analytical sample includes children born between 1981 and 1985 to women who were interviewed in Madhya Pradesh and who lived within 250 km of Bhopal (n=1260 births). The analytical sample for the analysis of cancer incidence includes men and women who were interviewed in Madhya Pradesh and were born between 1960 and 1990 (n=7031 men and 40 786 women).

A second dataset that we use is the Integrated Public Use Microdata Series from India for the year 1999, which presents harmonised data from the 1999 Socio-Economic Survey conducted by the National Sample Survey Organization (NSSO).[17] These data contain employment information for a representative sample, but only identify the district of the respondent. This is the most recent NSSO survey in which respondents were asked how long they had been living at their current residence, a crucial question for this analysis. The survey sample includes people from ages 6 to 64 years, thus including those who were in utero at the time of the BGD. The analytical sample for this analysis includes men interviewed in Madhya Pradesh and born between 1960 and 1990 (n=13 369).

The unit of observation in all analyses is the individual. The DHS/NFHS-4 and NSSO datasets could not be merged at the individual level, as the same respondents were not interviewed in each survey. In order to preserve the more precise geolocation of the NFHS-4, the two datasets were analysed separately.

### Patient and public involvement

Since DHS data are anonymised, direct involvement with study subjects is not possible.

### Measures

For the analysis of children born to women surveyed by the NFHS-4, the outcome of interest was the ratio of male-to-female births.

For men, the outcomes of interest were cancer rates and educational attainment (from the NFHS-4) and employment disability (from the NSSO). Cancer was assessed as a general question 'Do you currently have cancer', and educational attainment was assessed as the number of years that an individual attended school. Of the sample of men, 17 out of 7031 with non-missing response reported to have cancer, and the average number of years of education was 7.0 with an SD of 5.1. We use the question on whether 'the respondent was economically inactive because of disabilities or… other health-related reasons'

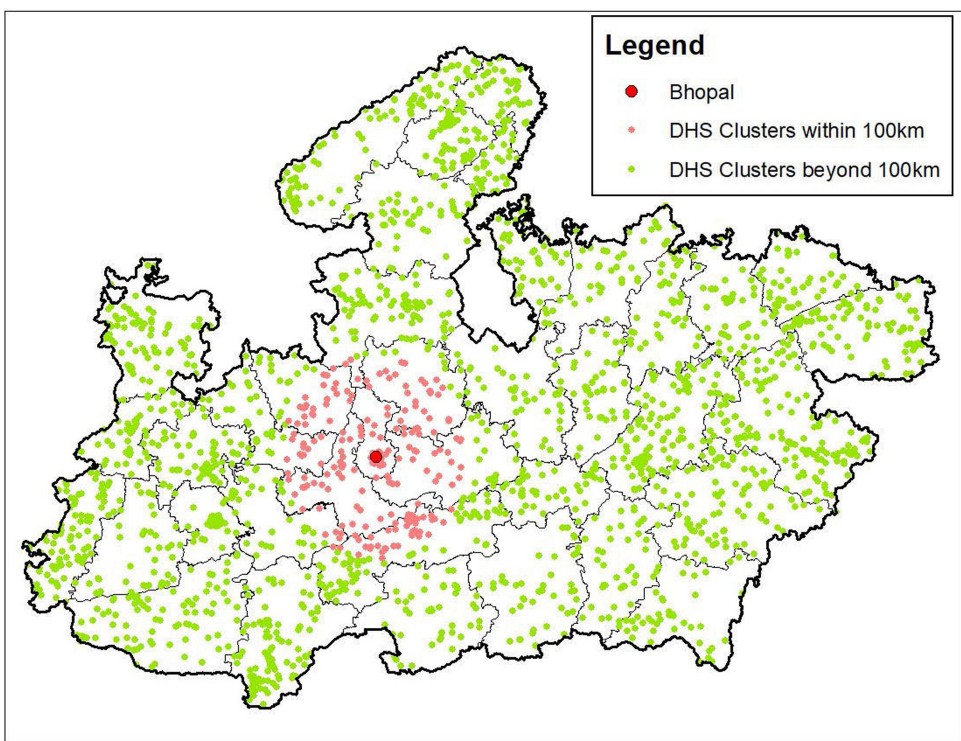

**Figure 1** Madhya Pradesh and DHS cluster locations around Bhopal. DHS, Demographic and Health Surveys.

to classify the person as suffering from employment disability. Forty-six men in the sample of 13 369 report to be suffering from employment disability. Given the prevalence of child labour, note that the employment-related questions of the survey are asked of all respondents, including children. Analyses assessing *movers* versus *non-movers* are necessarily restricted to men, since patrilocality in India means that most women move to their husband's house after marriage.[18] As 78% of Indian women aged 15–49 years have married,[16] the sample size of non-mover females is both small and non-representative, and thus cannot be considered in this analysis.

For all analyses, distance from the Union Carbide plant was the primary predictor. The NFHS-4 collected the latitude and longitude of each respondent's sampling cluster (henceforth, DHS clusters). The DHS clusters are villages or hamlets in rural areas and blocks or census tracts in urban areas; thus, they are much smaller geographical units than administrative units such as districts. These coordinates enable the calculation of distance from the Union Carbide plant in Bhopal for each respondent, within a privacy displacement radius of 2 km (urban) or 5–10 km (rural) (figure 1).

In the case of the district-level NSSO data, we measure distance from the Union Carbide plant to the nearest border of the district. For both datasets, we use the distance from the plant to form the treatment and control groups, since exposure to the toxic MIC gas would be more muted at greater distances.

## Analysis

The analysis uses a spatial difference-in-differences strategy, where distance from the Union Carbide plant

(we also refer to this as 'distance from Bhopal') lends the spatial dimension. We compare people living close to Bhopal with those far away from Bhopal, and people who were in utero at the time of the BGD with those who were older or not yet conceived at the time of BGD. Under some standard assumptions,[19] these estimates net out any difference in outcomes that arise due to *time-invariant* aspects of being close to Bhopal, as well as aspects of belonging to a specific birth cohort. The estimation also nets out the effect of other programmes that might have been occurring during these years across multiple birth cohorts, such as the universal immunisation programme.

All analyses were conducted using birth year cohorts (figure 2). The reference cohort includes those born between 1960 and 1974 (ie, respondents who were more than 10 years old at the time of the BGD). Three other cohorts are considered: young children below 10 years old at the time of the BGD (ie, respondents born between 1975 and 1984), fetuses (ie, respondents born in 1985) and children not yet conceived at the time of BGD (ie, respondents born between 1986 and 1990). This last cohort can be considered a placebo test, as we expect effects of the BGD on children not yet conceived to be null or significantly smaller than for cohorts that were alive or in utero.

It is worth noting that all four cohorts have been affected by the disaster, including our reference cohort. Therefore, while older cohorts are being used as a reference category, our results speak to the *relative* impacts on those who were in utero at the time of the disaster.

Linear regression models were used to estimate the relationship between proximity to Bhopal, birth cohort

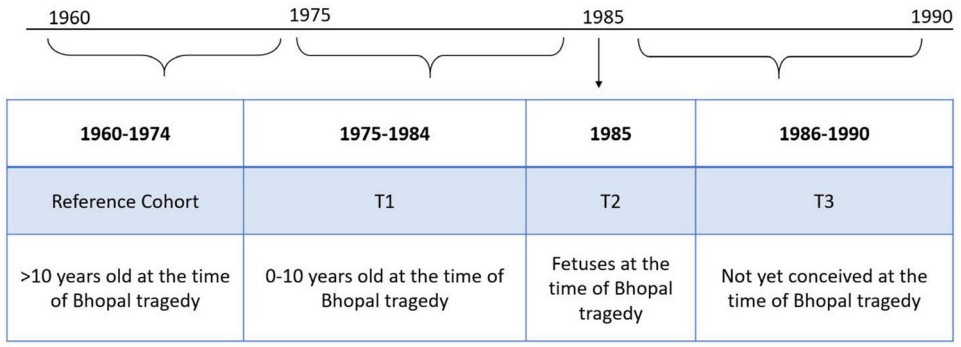

**Figure 2** Description of all cohorts used in empirical estimations.

and outcomes of interest, with the basic specification as follows:

(I)   $Y_{itc}=\alpha_1 Bhopal_c+\beta_1 \times T1_t \times Bhopal_c+\beta_2 \times T2_t \times Bhopal_c+\beta_3 \times T3_t \times Bhopal_c+Cohort\ FE+\varepsilon_{itc}$

where $Y_{itc}$ is the outcome of interest for person $i$ born in year $t$ and living in location $c$. We consider three health and education outcomes, namely: the probability of suffering from cancer, the probability of suffering from employment disability and years of education completed. $Bhopal_c$ takes the value of 1 if the respondent $i$ is living near Bhopal (we define 'near' Bhopal as being within 100 km following spatial patterns in sex ratios explained in figure 3) and 0 otherwise. $T1_t$ is a dummy variable equal to 1 if the respondent was born between 1975 and 1984 (cohort of ages 0–10 years at the time of BGD) and 0 otherwise. $T2_t$ is a dummy variable equal to 1 if the respondent was born in 1985 (in utero cohort) and 0 otherwise. $T3_t$ is a dummy variable equal to 1 if the respondent was born between 1986 and 1990 (cohort not yet conceived at the time of the BGD) and 0 otherwise. The omitted category consists of respondents who were older than 10 years at the time of the disaster and are living more than 100 km away from Bhopal. Cohort FE refers to birth year fixed effects to standardise the outcome variable across ages. SEs are two-way clustered by district and birth year in order to account for cross-sectional and serial autocorrelation within district, as well as covariate shocks across all people in the same birth cohort (such as changes in food prices).

$\beta_1$, $\beta_2$ and $\beta_3$, respectively, capture the effect of the BGD on young children below 10 years old, fetuses and not-yet-conceived children relative to those who were older than 10 years at the time of the BGD. The spatial difference-in-differences compares children in a cohort living near Bhopal with children in the same cohort living far from Bhopal. In order to interpret $\beta_1$, $\beta_2$ and $\beta_3$ as the causal effect of the BGD, the underlying identifying assumption is that in the absence of the disaster, the cohorts near and far from Bhopal would have evolved similarly. Note that $\alpha_1$ captures the effect of living near Bhopal vis-a-vis far from Bhopal for the reference cohort (1965–1974). This difference captures time-invariant differences in the two geographical locations.

## RESULTS

### Sex ratio changes following the BGD

The sex ratio of children born between 1981 and 1984 is markedly different from that of children born in 1985 (figure 3). Women who lived within 100 km of Bhopal experienced a relative decrease in the birth of males in the 1985 cohort (64% of children born from 1981 to 1984 were male, a proportion that drops to 60% in 1985). This is consistent with male fetuses more affected by external stress.[20–22] Women living beyond 100 km had no difference in the sex ratio across the 1981–1984 and the 1985 cohorts.

### Adult cancer rates among exposed children

Similar to patterns seen with changes in the sex ratio, children who were born in 1985 (the year following the BGD) and who lived close to Bhopal during the disaster experience much higher rates of cancer as adults compared with adults who were born before or after the disaster and who lived further away from Bhopal (figure 4). As in the case of differences in sex ratios, the abnormally high cancer rates appear to extend up to 100 km from Bhopal.

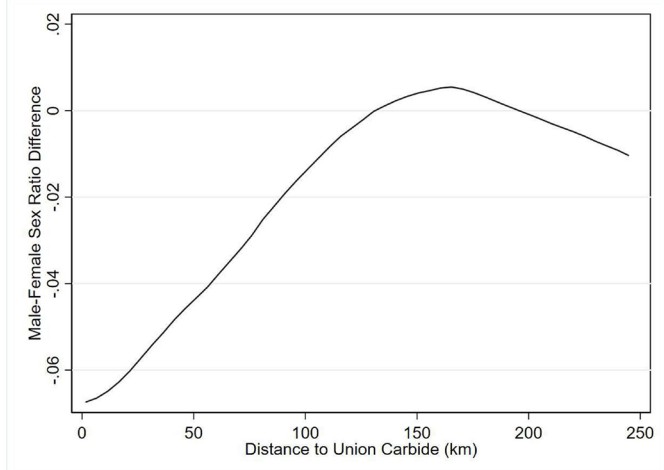

**Figure 3** Difference in male–female sex ratio between 1985 and 1981–1984 cohorts among children born to women aged 15–49 years in Madhya Pradesh, by distance to Union Carbide plant.

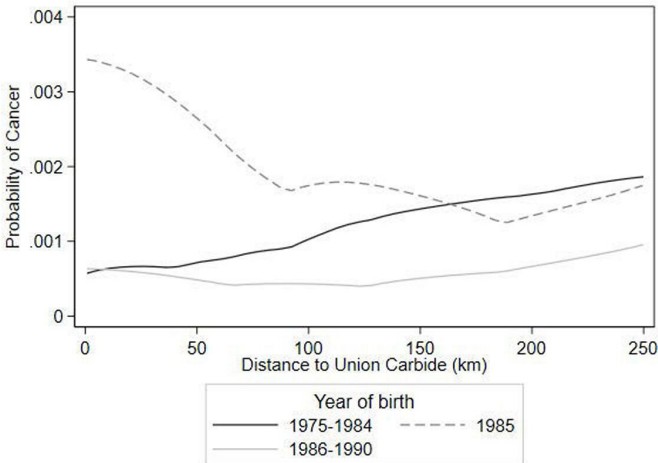

**Figure 4** Self-reported cancer incidence among men and women living in Madhya Pradesh in 2015, by distance to Union Carbide plant.

Given the two spatial patterns in figures 3 and 4, we use 100 km as the BGD exposure cut-off in our analysis.

Among all men in the NFHS-4 analytical sample, estimates suggest that the 1975–1984 and 1986–1990 birth cohorts have statistically indistinguishable cancer rates compared with the 1960–1974 cohort, while men born within 100 km of Bhopal in 1985 have a 2.1 percentage point higher risk than the other cohorts (table 1). This is

**Table 1** Probability of suffering from cancer among men born in Madhya Pradesh

| | Prob (reports having cancer) | | |
| --- | --- | --- | --- |
| | (1) Pooled | (2) Non-movers | (3) Movers |
| 1986–1990 | −0.00373 | 0.00170 | −0.0141 |
| | (0.00340) | (0.00147) | (0.0121) |
| 1985 | 0.0209*** | 0.0624*** | −0.00906 |
| | (0.00591) | (0.00505) | (0.0113) |
| 1975–1984 | −0.00459 | −0.00157 | −0.00906 |
| | (0.00355) | (0.00181) | (0.0113) |
| 1960–1974 | Reference | Reference | Reference |
| Observations | 7031 | 5002 | 2029 |
| $R^2$ | 0.007 | 0.012 | 0.014 |
| Cohort FEs | Yes | Yes | Yes |
| Control mean | 0.00252 | 0.00228 | 0.00321 |

The cohort of men born in 1985 within 100 km of Bhopal has a higher likelihood of cancer compared with other cohorts and to those born in Madhya Pradesh beyond 100 km from Bhopal (1), with effect most pronounced among men who have never moved (2) instead of those who have moved (3). Estimates are from ordinary least squares regressions as in equation (1) with controls consisting of FEs for year of birth cohort and whether the DHS cluster is within 100 km of Bhopal. SEs are two-way clustered at the district and year of birth levels.
Robust SEs in parentheses.
***P<0.01, **p<0.05, *p<0.1.
DHS, Demographic and Health Surveys; FEs, fixed effects.

an eightfold higher cancer risk compared with the other cohorts.

Among men who have never moved, and were therefore residing within 100 km of the Union Carbide plant during the BGD, the effect of being born in 1985 and near Bhopal is even stronger. These men have a 6.2 percentage point higher cancer risk in 2015 compared with the other cohorts and with those living more than 100 km from Bhopal (Given the presence of both fixed effects and interaction terms in the difference-in-differences framework, the linear probability model estimated with ordinary least squares is more appropriate than logistic regression. For completeness, we report the OR for the 1985 cohort from a conditional logic model: 3.36 (0.1–111.8). While the estimation problems with logistic in the difference-in-differences context mean we cannot interpret the magnitude, the sign is consistent with the results we report from ordinary least squares.). This represents a 27-fold higher risk of cancer among adults who were in utero during the BGD, suggesting long-term health consequence of exposure, even while in the womb. Men who moved at some point before the survey in 2015/2016 (and who were therefore less likely to be exposed to the BGD) had no difference in cancer rates across cohorts.

### Effects on employment disability
Men who were in utero during the BGD and who lived in districts with a border within 100 km of Bhopal are 1 percentage point more likely to report employment disability when they are surveyed as adults, compared with the older cohorts and those living further from Bhopal (figure 5 and online supplemental table 1). The effect increases to 2 percentage points among those living within 50 km of Bhopal (online supplemental table 2), consistent with higher intensity of exposure to MIC among those living closer to Bhopal.

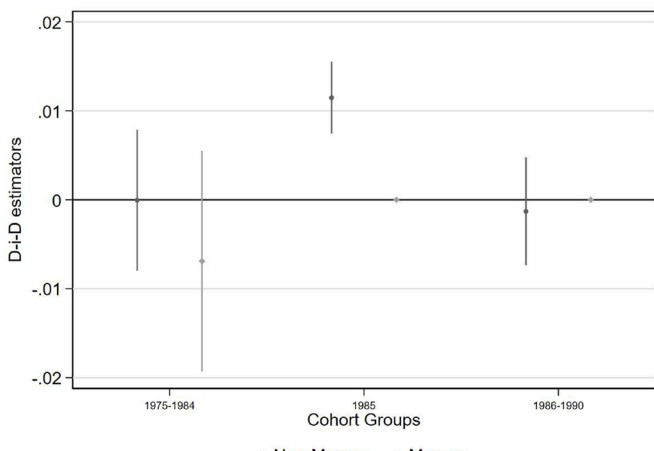

**Figure 5** Employment disability, compared with reference group of men born from 1960 to 1974 and living >100 km from Bhopal. Bars represent 95% CIs. Note that there were no individuals reporting employment disability in the 1985 and 1986–1990 cohorts among the *movers* subgroup. D-i-D, difference-in-differences.

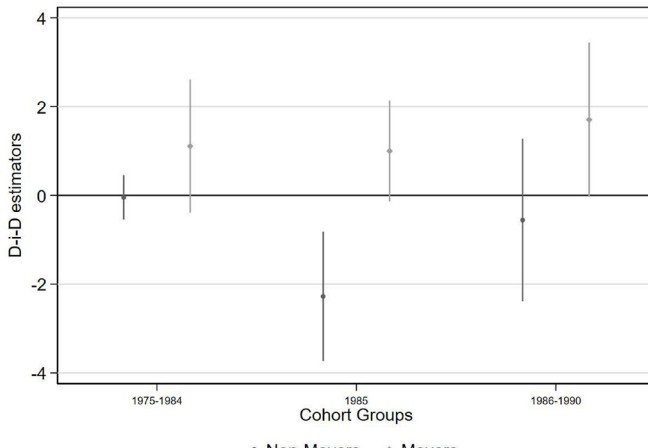

**Figure 6** Years of education completed, compared with reference group of men born from 1960 to 1974 and living >100 km from Bhopal. D-i-D, difference-in-differences.

### Effects on educational attainment

Relative to other cohorts, men who were in utero during the BGD and who lived in districts with a border within 100 km of Bhopal attained approximately 2 fewer years of education, with the effects only evident among men who have never moved (figure 6 and online supplemental table 3). Limiting the treatment radius to 75 km yields the same estimated effect, while smaller distances yield no discernible effect (online supplemental table 4).

### DISCUSSION

The BGD was one of the most serious industrial accidents in history, causing thousands of deaths at the time of the leak, and tens of thousands more deaths and serious health sequelae in the three decades since.[23 24] This analysis documents long-term, intergenerational impacts of the BGD, showing that men who were in utero at the time of the BGD were more likely to have a disability that affected their employment 15 years later, and had higher rates of cancer and lower educational attainment over 30 years later. Importantly, these results indicate social costs stemming from the BGD that extend far beyond the mortality and morbidity experienced in the immediate aftermath. Moreover, our results suggest that the BGD affected people across a substantially more widespread area than has previously been demonstrated: up to around 100 km (approximately 60 miles) from the site, as opposed to the 4.5 km radius that was considered exposed by public health officials and researchers after the disaster.[9 10 14 25] This study is also the first, to our knowledge, to use different birth cohorts to attribute causality of subsequent health and social consequences to the BGD.

We find that men currently living within 100 km of Bhopal and born in 1985 have an eightfold higher risk of cancer than men of other birth cohorts; of those men, those who have never changed residence since the BGD have a 27-fold higher risk of cancer. While elevated cancer levels have been previously associated with MIC exposure,[14 26–28] our results identify long-term, birth year-based differences in cancer prevalence. This underscores the plausibility of multiple MIC exposure pathways, including teratogenic in utero exposure, and ongoing environmental exposure. Bhopal has ongoing cancer surveillance programmes,[29] and it remains clear that those exposed to MIC require ongoing and careful monitoring, support and treatment.

Employment disability was 1 percentage point more likely among men who were in utero in districts within 100 km of Bhopal during the BGD than those born prior or more distal to the BGD. This is a meaningful impact since baseline employment disability rates are quite low (0.4%), and men's employment at the time of study was nearly universal at 98%.[16] In a context of high poverty, with nearly 40% of Madhya Pradesh's population living beneath the poverty line,[30] more than 90% informal employment nationally (thus lacking social protection and safety nets),[31] and household reliance on male income due to low levels of female labour force participation and a highly skewed gender wage gap,[16 32] this increase in disability or health-related economic inactivity risks substantial impact on the health, safety and well-being of affected families.

The finding that men who were in utero and within 100 km of Bhopal during the BGD received more than 2 fewer years of education than other cohorts has tremendous human capital implications. This is a large impact since the average number of years of education in the control group is only 5.6 years, and because education has such a direct association with subsequent wages and consumption.[33] We cannot ascertain whether the lower educational attainment is due to health or cognitive consequences of being exposed to the BGD in utero. Given that the BGD effect is much more evident in the 1985 cohort compared with older or younger children, it is unlikely that the effect operated through broader socioeconomic impacts or impacts on other family members that limited children's access to school. Regardless of the mechanism, the reduced educational attainment is evidence of long-term consequences of exposure to the accident that imply personal and social costs far beyond the immediate health impacts.

A few study limitations are worth outlining. First, as with any ecological exposure assignment, the cohort of in utero children we assign to exposure will include a range of actual exposure to MIC gas. The interpretation of our estimates would be the effect of average exposure in the population we define as treated. Second, our estimates are affected by mortality and migration. An important consequence of a disaster, whether natural or man-made, is that mortality can severely affect the composition of the population observed post-facto. In our empirical context, any mortality effect of the disaster that symmetrically affects all cohorts within Bhopal (without affecting people away from Bhopal) will not affect our estimates. As such, it is differential mortality across cohorts due to the BGD that

would have compositional effects on our study sample. For example, if the weakest children in utero were more likely to die from the incident, then our estimates could be considered a lower bound.

Likewise, a frequent human response to a disaster is migration to safer areas. Again, it would only be differential migration in response to the BGD (different migration across cohorts and across people living close or far from Bhopal) that would create a problematic compositional difference between our treated cohort and the others. In fact, migration in this area was relatively low and researchers found that 91% of the population remained in the same area subsequent to 1984.[10] Our own estimates in online supplemental figure 1 suggest that mothers of the in utero cohort are not significantly different from mothers of other consecutive cohorts on various educational and health dimensions.

It is also important to note that the long-term consequences that we estimate could be the result of both direct effects from exposure as well as lack of subsequent mitigation of the effects through health, disability and education services. While disentangling these forces is important from a policy perspective, in this paper, we simply highlight the total combined effect of being exposed to such disasters as a child. As such, causal inference methods mitigate but do not eliminate the need to consider different routes of causation.

Work in other empirical contexts of industrial disasters should strive to estimate how effects on in utero children vary by trimester of pregnancy. Other kinds of shocks have been shown to vary in their effect by trimester[34]; however, given data quality issues on long recall of month of birth in the DHS,[35] we do not attempt to estimate effects by trimester in this study.

Finally, cancer reporting in the data comes from self-reports, which could be subject to biases; there are also very few cases of self-reported cancer in the data. We think of our results on cancer as being consistent with the broader public health narrative on the consequences of the disaster, but more data through systematic cancer screening and place of birth information are key to firmly establish this link.

Understanding the short-term and long-term damages caused by industrial disasters is key to gaining insight into the trade-offs involved in making regulatory decisions. It is also crucial from a policy response perspective if policymakers wish to dedicate resources to mitigate harm done by such events or to compute legal damages. These concerns are particularly germane now, when there is evidence that the multiple health conditions suffered by many BGD survivors may make them more susceptible to the COVID-19 pandemic.[36 37] The evidence presented in this paper starkly highlights the long-term, intergenerational health and human capital effects of the BGD, and underscores the need for ongoing survivor support, as well as robust regulatory protection.

**Contributors** PB conceptualised the project. GCM and AK organised data and conducted analysis. All authors including LM and AR contributed equally to data interpretation, writing, review and editing. GCM verified the underlying data and is guarantor.

**Funding** AR and LM received funding from the Bill and Melinda Gates Foundation (INV-008648).

**Map disclaimer** The inclusion of any map (including the depiction of any boundaries therein), or of any geographic or locational reference, does not imply the expression of any opinion whatsoever on the part of BMJ concerning the legal status of any country, territory, jurisdiction or area or of its authorities. Any such expression remains solely that of the relevant source and is not endorsed by BMJ. Maps are provided without any warranty of any kind, either express or implied.

**Competing interests** None declared.

**Patient and public involvement** Patients and/or the public were not involved in the design, or conduct, or reporting, or dissemination plans of this research.

**Patient consent for publication** Not required.

**Ethics approval** All data used in this study are publicly available and de-identified. Ethical approval for NFHS-4 data collection was provided by the International Institute for Population Sciences and ICF. Ethical approval for the 1999 NSSO Socio-Economic Survey was provided by the Indian National Sample Survey Office. Ethical exemption for this analysis was provided by the University of California San Diego IRB.

**Provenance and peer review** Not commissioned; externally peer reviewed.

**Data availability statement** Replication code will be made available upon publication. The raw DHS survey data are subject to a user agreement and are available from the Demographics and Health Surveys Program at https://dhsprogram.com. The IPUMS data with the 1999 National Sample Survey are available from https://international.ipums.org/international/index.shtml.

**ORCID iDs**
Gordon C McCord http://orcid.org/0000-0001-5042-9225
Lotus McDougal http://orcid.org/0000-0002-3002-0489

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
