## [Reviewer comments · BMJ Open]

This paper was submitted to a another journal from BMJ but declined for publication following peer review. The authors addressed the reviewers' comments and submitted the revised paper to BMJ Open. The paper was subsequently accepted for publication at BMJ Open.

ARTICLE DETAILS

TITLE (PROVISIONAL)	Long-term Health and Human Capital Effects of In Utero Exposure to an Industrial Disaster: A Spatial Difference-in-Difference Analysis of the Bhopal Gas Tragedy
AUTHORS	McCord, Gordon C; Bharadwaj, Prashant; McDougal, Lotus; Kaushik, Arushi; Raj, Anita

VERSION 1 – REVIEW

REVIEWER	Sarwa, Khomendra Government Polytechnic College for Girls, Pharmacy
REVIEW RETURNED	30-Sep-2022

GENERAL COMMENTS	In this review the objective of research study is clearly defined. The study design was also good, and I agree with both the results and the conclusion. Some suggestions are below for this manuscript: This work looks like interesting and worth publishing. The topic of the manuscript is suitable for work. The manuscript is clearly presented and well organized: The manuscript gives adequate references to related work. The English in the manuscript is satisfactory. However, Although the manuscript is well written, there are a few grammatical/typographical errors that need to be corrected. In many places, there is no uniformity in writing like in utero (in-utero), long-term etc. Special attention required by author in highlighted words in the manuscript which is attached.
--

REVIEWER	Prakash, Nishith Northeastern University College of Social Science and Humanities, School of Public Policy and Urban Affairs
REVIEW RETURNED	30-Oct-2022

GENERAL COMMENTS	This is a terrific paper, and I am surprised that no one had studied the impact of the Bhopal Gas Disaster, which was one the most serious industrial accidents in history, causing thousands of deaths at the time of the leak, and tens of thousands more deaths and serious health sequelae in the three decades since. The authors have performed careful analysis and the results is relevant for public policy.
--

	I have few comments:  1. Conceptually I will like the authors to classify outcomes as primary and secondary (based on literatures). This is especially relevant for education outcomes – as a reader I would like to know the underlying mechanism. My guess is its mediated via cognitive ability and not say supply of schools. 2. Additional outcomes:  a. Migration: Did households move after this? This also helps as the authors don't need to control for migration. b. Outcomes for females: I understand the focus on male, but I will be very interested to see how this worsened gender gaps. 3. Vary intensity: I believe by distance cut-offs; the authors can vary the intensity of the impact (although the sample size will reduce drastically). 4. In-utero exposure: I believe the impact will vary depending how long is a kid in-utero. Suppose someone is one month old vs. close to delivery – will this matter? 5. Tables: Please add detailed footnotes and expand the set of controls. 6. I was expecting to see some placebo results.
--	--

REVIEWER	Kumar, Santosh University of Notre Dame, Keough School of Global Affairs
REVIEW RETURNED	02-Nov-2022

GENERAL COMMENTS	Summary: Analyzing micro-level data (NFHS and NSSO), the authors estimate the long-term effects of in-utero exposure to the Bhopal Gas Disaster (BGD) on disability, cancer incidence, and educational attainment in India. The authors use the spatial difference-in-difference method to estimate the causal association between BGD and the outcome variables. Cohorts exposed to BGD in-utero had higher disability and cancer rates and lower educational attainment 30 years later. It is a well-done study and I enjoyed reading it. Below are the points that the authors should consider in the revised manuscript. Main points:  1. Page 3- Introduction- “Our findings offer timely.....” is a strong policy conclusion, particularly rollbacks of policies. Consider toning down this a bit. 2. Introduction, page 2: Authors mention that exposure to toxic gas led to increase in the rate of miscarriage, stillbirths, or neonatal mortality. This may change the composition of the surviving pool of children and this biases the results. The authors discuss the issue of selective mortality in the discussion. Is it possible that BGD may have affected cohorts differentially (may be age- younger kids or kids living closer to the accident site may be more likely to die compared to older kids and kids living far off from the accident site?) 3. One limitation of the paper is that it does not contain information on birthweight or gestational weeks- in-utero shocks are likely to affect birth outcomes- this is acknowledged in the conclusions but what is missing is the discussion of consequences of omitting the birthweight variable from the analysis.
---

	4. The paper uses data from Madhya Pradesh for the analysis however it is unclear whether the analytical sample is representative of the country. How would this limitation influence the outcomes and policy recommendations needs to be clearly outlined in the paper? 5. The in-utero cohort (1985 cohort) was exposed to the BGD for 9 months. Epidemiology or biomedical literature indicates that the effects of in-utero shocks differ by trimester. Do authors have data to analyze the effects by trimester? There could be a power issue but it is worth discussing the implication of this in the discussion section. 6. Eq 1 needs elaboration and should be clearly spelled out. Are cohort FE a dummy for T1, T2, and T3 and there are additional terms in the equation controlling for year of birth? It is common to include the main effects when the model has interaction terms. 7. Contemporaneous program- One issue with the DiD model is that there should not be any contemporaneous programs at the same time. I recall Universal Immunization Program (UIP) was rolled out in 1985 across districts in India between 1985-90. Could authors discuss the implications of this for their main findings?
--	---

REVIEWER	Waller , Lance A. Rollins School of Public Health, Biostatistics and Bioinformatics
REVIEW RETURNED	20-Dec-2022

GENERAL COMMENTS	General Comments 1. Many thanks to the authors for a carefully conducted study and for a thoughtfully written manuscript. 2. The authors use linear regression where the outcome is the probability of certain health outcomes. In linear regression studies of the probability of health outcomes (particularly rare outcomes) it is more typical to use logistic regression. When logistic regression is not used to model rare outcomes (and I understand that the difference-in-differences approach is more commonly applied to linear regressions), most analysts use the natural logarithm of the observed risk to move errors closer to normal distributions. Did the authors try their approach on the log-probabilities? I suspect the results may be quite similar but it would be worth noting so. Specific Comments (line numbers refer to actual lines of text, not the lines in the margin) 1. Page 2, line -3. I suggest shifting “adverse health and social impacts” to “adverse health and social impacts of the BGD event”. 2. Page 3, Data section, first paragraph line 7. “n=1,260 births with nonmissing...”. What was the total number of births reported (so the 1,260 number can be put in context). 3. Page 4, line -8. “under some standard assumptions”. It would be helpful to give a general reference to difference-in-differences methods here for readers interested in more details on the standard assumptions. 4. Page 4, line -3. “below 10 years old” to “below 10 years old at the time of BGD”.
--

	5. Page 5, last line of paragraph containing equation 1. “Standard errors...to account for spatial and serial autocorrelation”. I have two quick questions regarding the standard error calculations: a. Are the standard errors also adjusted for the population size in each district? The variance of estimated disease risks, especially for rare outcomes, is dependent on the number of individuals observed. b. I believe that the clustering by birth year and district will adjust for observations with the same district-year. In a sense, this adjusts for “spatial autocorrelation” by allowing repeated-measures-type correlations between observations from the same spatial region, but (I don’t believe) this is not the same as adjusting for the more typical use of the term “spatial autocorrelation” where observations are correlated to other nearby observations where nearby can mean in adjacent regions or with correlations decreasing as distance between observations increase. If this is the case, I suggest the authors adjust “for spatial and serial autocorrelation” to “for correlations between observations occurring in the same year and district”. 6. Page 6, line 2. “born in 1985” to “born in 1985 (the year following BGD)”. 7. Page 6, Table 1. Are the results in Table 1 limited to those living within 100 km at the time of BGD? The text suggests this is the case, but the table caption suggests the results apply to all men born in Madhya Pradesh. 8. Page 7-8. Thanks to the authors for a thoughtful discussion. Woven in the discussion of potential environmental and social drivers of risk, is an interesting point about the type of causal conclusions enabled by the causal inference approach. The results provide causal insight to the impact of the BGD *event* itself (time and location), but, as noted by the authors, the results do not provide causal insight into the impact of the MIC exposures themselves. In the expansion of the application of causal inference within epidemiology, I feel this is an important point to specifically articulate. The authors’ analysis provides insight into the BGD event and the types of outcomes detail some of the broad-reaching impacts of the event on health and social impacts. The paragraph on page 8 beginning “A few study limitations...” is insightful in bringing interpretation to elements of the event. To bring this home, I suggest adding a summary statement at the end noting that the use of causal inference mitigates but does not eliminate the need to consider different routes of causation. I found the discussion here to be very helpful in bringing to light some nagging concerns regarding the need for care in interpretation of causal inference methods that I had not quite been able to articulate prior to reading the manuscript. Well done.
--	---

VERSION 1 – AUTHOR RESPONSE

Reviewer: 1

Dr. Khomendra Sarwa, Government Polytechnic College for Girls, Government Girls poly

Comments to the Author:

in this review the objective of research study is clearly defined. The study design was also good, and I agree with both the results and the conclusion.

Some suggestions are below for this manuscript:

This work looks like interesting and worth publishing.

The topic of the manuscript is suitable for work.

The manuscript is clearly presented and well organized:

The manuscript gives adequate references to related work.

The English in the manuscript is satisfactory.

However, Although the manuscript is well written, there are a few grammatical/typographical errors that need to be corrected.

Thank you for this comment. We have now gone through and done our best to correct these.

In many places, there is no uniformity in writing like in utero (in- utero), long-term etc.

Special attention required by author in highlighted words in the manuscript which is attached.

Thank you for pointing these out. We have made these uniform in the updated version and made edits to the highlighted words as appropriate.

Reviewer: 2

Dr. Nishith Prakash, University of Connecticut

Comments to the Author:

This is a terrific paper, and I am surprised that no one had studied the impact of the Bhopal Gas Disaster, which was one the most serious industrial accidents in history, causing thousands of deaths at the time of the leak, and tens of thousands more deaths and serious health sequelae in the three decades since. The authors have performed careful analysis and the results is relevant for public policy.

I have a few comments:

1. Conceptually I will like the authors to classify outcomes as primary and secondary (based on literatures). This is especially relevant for education outcomes – as a reader I would like to know the underlying mechanism. My guess is its mediated via cognitive ability and not say supply of schools.

Thank you for this comment. We agree that distinguishing primary and secondary outcomes makes sense in this context. For this paper, we do not study the first-order primary outcomes (deaths and disease in the immediate aftermath of the accident) and instead study only long-term effects. Among these, it is reasonable to consider health effect the primary outcomes, which we study in Figures 3, 4 and Table 1. This is also supported by the numerous public health papers on short-term and long-term BGD health impacts that we cite (references 7-10, 12-14). We then make it clear that subsequent effects on education are the secondary outcomes. We also cite the relevant literature linking early life health to long term educational attainment (reference 15).

With regards to mechanism for education outcomes, we note in the paper that we cannot pinpoint exact mechanism (“cannot ascertain whether the lower educational attainment is due to health or cognitive consequences of being exposed to the BGD in utero”) nevertheless we note the fact that impacts are observed on a single cohort make it “unlikely that the effect operated through broader socioeconomic impacts or impacts on other family members that limited children’s access to school.” As such, we agree with the referee that the mechanism is not through school supply.

2. Additional outcomes:

- a. Migration: Did households move after this? This also helps as the authors don’t need to control for migration.

This is an important point, as it is reasonable to believe that people moved after this incident. However, migration in this area was relatively low and researchers have found that 91% of the population remained in the same area subsequent to 1984 (Dhara & Dhara [2002], reference 10). Nevertheless, we consider it important in our empirical exercise to show the results for those who have always stayed in their place of birth compared to people who have not (columns 2 vs 3 in Tables 1, S1 & S2; and coefficient plots in Figures 5 & 6 distinguishing non-movers vs. movers).

- b. Outcomes for females: I understand the focus on male, but I will be very interested to see how this worsened gender gaps.

Given your point above about migration, this is especially hard to do given that women almost always out-migrate upon marriage in India (Khalil & Mookerjee [2019], reference 18). Not knowing from the DHS where these women were born, assigning BGD exposure based on location at time of interview creates a large potential for treatment misassignment. In addition, the 20% of women who have never moved are not likely to be representative of the entire female population. While we agree that gender gaps are key to understand in this context, given these data limitations, we necessarily focus this study on the male population.

3. Vary intensity: I believe by distance cut-offs; the authors can vary the intensity of the impact (although the sample size will reduce drastically).

The referee’s thinking is correct, and the non-parametric analysis in Figures 3 & 4 reflect the distance decay function of effects and suggest a maximum effect radius of 100 km. The sample size issues mentioned by the referee limit our ability to thoroughly measure impact heterogeneity across distance

bands. To give a sense, however, below are tables showing how the effect on employment disability and educational attainment vary when we vary the distance cutoff. As the referee predicted, closer to Bhopal we measure larger effect sizes on employment disability (this pattern is not evident for years of schooling completed). Note that since we expect places to be impacted up to 100 km, when restricting the radius we omit observations between the treatment radius and 100 km such that the control group is always beyond 100 km.

We have added these tables to the supplementary material, and mentioned this in the text. We thank the referee for the suggestion.

Prob(Employment Disability) among Non-Movers, by Treatment
Radius

	(1)	(2)	(3)
	100 km	75 km	50 km
1986-1990	-0.00131	-0.00183	-0.00045
	(0.00297)	(0.00390)	(0.00445)
1985	0.0115***	0.0138***	0.0203**
	(0.00198)	(0.00217)	(0.0078)
1975-1984	-4.65e-05	-0.00049	-0.00004
	(0.00388)	(0.00314)	(0.00294)
1960-1974	Reference	Reference	Reference
Observations	12,129	11,799	11,100
R-squared	0.003	0.002	0.003
Cohort FEs	Yes	Yes	Yes

Control Mean	0.00426	0.00437	0.0043
--------------	---------	---------	--------

Robust standard errors in parentheses

*** p<0.01, ** p<0.05, * p<0.1

Years of Education Completed among Non-Movers, by Treatment Radius

	(1)	(2)	(3)
	100 km	75 km	50 km
1986-1990	-0.557 (0.897)	-0.356 (0.682)	-0.264 (0.454)
1985	-2.276*** (0.714)	-2.253** (0.983)	-0.316 (2.200)
1975-1984	-0.0471 (0.245)	-0.371 (1.292)	-0.900 (1.970)
1960-1974	Reference	Reference	Reference

Observations	5,014	4,830	4,702
R-squared	0.055	0.055	0.053
Cohort FEs	Yes	Yes	Yes
Control Mean	5.561	5.584	5.643

Robust standard errors in parentheses

*** p<0.01, ** p<0.05, * p<0.1

4. In-utero exposure: I believe the impact will vary depending how long is a kid in-utero. Suppose someone is one month old vs. close to delivery – will this matter.

This is a good question. Presumably this should matter given the literature on trimester level shocks and their impacts (Almond and Mazumder 2011). However, we note that recall of birth month data in the DHS is often incorrect (Larsen et al 2019), because of which empirical exercises requiring long recall on month of birth are ill-advised.

Larsen, A. F., Headey, D., & Masters, W. A. (2019). Misreporting month of birth: Diagnosis and implications for research on nutrition and early childhood in developing countries. *Demography*, 56 (2), 707-728.

Almond, Douglas, and Bhashkar Mazumder. (2011). "Health Capital and the Prenatal Environment: The Effect of Ramadan Observance during Pregnancy." *American Economic Journal: Applied Economics*, 3 (4): 56-85.

5. Tables: Please add detailed footnotes and expand the set of controls.

Thank you for pointing this out. We have now added detailed footnotes to Table 1 in the paper and Tables S1 & S2 in the supplemental material.

6. I was expecting to see some placebo results.

In a sense the post 1985 births are our placebo results. We might not have explicitly spelled it out this way, but we do now. Thank you for the suggestion.

Reviewer: 3

Dr. Santosh Kumar, University of Notre Dame

Comments to the Author:

Report

Title: In Utero Exposure to Industrial Disasters: A Case Study of the Bhopal Gas Tragedy

Manuscript ID: BMJ open-2022-066733

Summary:

Analyzing micro-level data (NFHS and NSSO), the authors estimate the long-term effects of in-utero exposure to the Bhopal Gas Disaster (BGD) on disability, cancer incidence, and educational attainment in India. The authors use the spatial difference-in-difference method to estimate the causal association between BGD and the outcome variables. Cohorts exposed to BGD in-utero had higher disability and cancer rates and lower educational attainment 30 years later. It is a well-done study and I enjoyed reading it. Below are the points that the authors should consider in the revised manuscript.

Main points:

1. Page 3- Introduction- "Our findings offer timely....." is a strong policy conclusion, particularly rollbacks of policies. Consider toning down this a bit.

We agree, and have rephrased the sentence as follows: "Our findings offer timely information to inform debate on potential rollbacks of policies related to social and environmental protections attached to industrialization in India and elsewhere."

2. Introduction, page 2: Authors mention that exposure to toxic gas led to increase in the rate of miscarriage, stillbirths, or neonatal mortality. This may change the composition of the surviving pool of children and this biases the results. The authors discuss the issue of selective mortality in the discussion. Is it possible that BGD may have affected cohorts differentially (may be age- younger kids or kids living closer to the accident site may be more likely to die compared to older kids and kids living far off from the accident site?)

We agree with the referee, and mention in the discussion that “it is differential mortality across cohorts due to the BGD that would have compositional effects on our study sample.” As we note in that same paragraph, however, if weaker in-utero children suffered mortality at higher rates than older children, then our estimates would be interpreted as a lower bound. We have moved the relevant sentence to the end of the paragraph to make the interpretation point clear.

3. One limitation of the paper is that it does not contain information on birthweight or gestational weeks- in-utero shocks are likely to affect birth outcomes- this is acknowledged in the conclusions but what is missing is the discussion of consequences of omitting the birthweight variable from the analysis.

While the focus of our study is on documenting long-term effects of exposure on the in utero cohort, we agree that documenting the effects of the disaster on birthweight and gestational weeks on the in utero cohort is important. However, the NFHS does not contain that information for interviewees who are now adults. While the literature does not include a documented effect of MIC on birth weight per se, we note that the literature has documented effects of exposure on fetal weight (Kanhere et al., 1987). The meta-studies we cite in the introduction include this finding, which we do not discuss further in the paper given our focus on long-term impacts.

Kanhere S, Darbari BS, Shrivastava AK. Morphological study of expectant mothers exposed to gas leak at Bhopal. *Indian J Med Res* 1987; 86(Suppl):77-82.

4. The paper uses data from Madhya Pradesh for the analysis however it is unclear whether the analytical sample is representative of the country. How would this limitation influence the outcomes and policy recommendations needs to be clearly outlined in the paper?

We thank the referee for the question. While the NFHS-4 is representative at the district level, our aim is not to measure population-level effects of BGD. Our research design identifies an internally valid causal estimate of the BGS on our outcome variables among the population in the sample. We do not reweight observations to calculate population-wide effects, as we are primarily interested in documenting the causal effect (Solon et al., 2013).

Solon, Gary, Steven J. Haider and Jeffrey M. Wooldridge, “What Are We Weighting For?” *Journal of Human Resources* 50(2): 301-317, 2015.

5. The in-utero cohort (1985 cohort) was exposed to the BGD for 9 months. Epidemiology or biomedical literature indicates that the effects of in-utero shocks differ by trimester. Do authors have data to analyze the effects by trimester? There could be a power issue but it is worth discussing the implication of this in the discussion section.

This is a good question. Presumably in utero shocks do vary by trimester given the literature on trimester level shocks and their impacts (Almond and Mazumder 2011). However, we note that recall of birth month data in the DHS is often incorrect (Larsen et al 2019), because of which empirical exercises requiring long recall on month of birth are ill-advised.

Larsen, A. F., Headey, D., & Masters, W. A. (2019). Misreporting month of birth: Diagnosis and implications for research on nutrition and early childhood in developing countries. *Demography*, 56 (2), 707-728.

Almond, Douglas, and Bhashkar Mazumder. (2011). "Health Capital and the Prenatal Environment: The Effect of Ramadan Observance during Pregnancy." *American Economic Journal: Applied Economics*, 3 (4): 56-85.

6. Eq 1 needs elaboration and should be clearly spelled out. Are cohort FE a dummy for T1, T2, and T3 and there are additional terms in the equation controlling for year of birth? It is common to include the main effects when the model has interaction terms.

We apologise for the lack of clarity and have updated the paragraph to clarify. The cohort FE is a dummy for the year of birth, which is why the main effects of T1, T2 and T3 are excluded (they are subsumed by the year of birth dummies).

7. Contemporaneous program- One issue with the DiD model is that there should not be any contemporaneous programs at the same time. I recall Universal Immunization Program (UIP) was rolled out in 1985 across districts in India between 1985-90. Could authors discuss the implications of this for their main findings?

The strength of the spatial DiD model is that any potentially confounding programs would have to be not only contemporaneous but differentiated geographically in the same way as our design. Since we find that cancer rates, employment disability and educational attainment are worse for the 1985 cohort than for earlier and later cohorts, it is highly unlikely that the UIP would have an effect only on a single cohort and only within 100 km of Bhopal. We have added a sentence to the methods section adding this explanation: " The estimation also nets out the effect of other programs that might have been occurring during these years across multiple birth cohorts, such as the Universal Immunization Program."

Reviewer: 4

Dr. Lance A. Waller , Rollins School of Public Health

Comments to the Author:

Comments on BMJ Open manuscript bmjopen-2022- "0066733 "In utero exposure to industrial disasters: A case study of the Bhopal Gas Tragedy".

General Comments

1. Many thanks to the authors for a carefully conducted study and for a thoughtfully written manuscript.

Thank you.

2. The authors use linear regression where the outcome is the probability of certain health outcomes. In linear regression studies of the probability of health outcomes (particularly rare outcomes) it is more typical to use logistic regression. When logistic regression is not used to model rare outcomes (and I understand that the difference-in-differences approach is more commonly applied to linear regressions), most analysts use the natural logarithm of the observed risk to move errors closer to normal distributions. Did the authors try their approach on the log-probabilities? I suspect the results may be quite similar but it would be worth noting so.

Since our observation-level outcomes for cancer and employment disability are binary variables, taking logs is infeasible. The referee is correct that logistic regression is more common with these kinds of outcomes, but in our case the employment of interaction terms to estimate difference-in-differences creates well-documented problems with logistic regression estimation (Ai and Norton, 2003). Moreover, the conditional logit needs to be used in order to avoid the incidental parameters problem of logistic estimation with fixed effects (Chamberlain, 1980).

We have added a footnote noting to the reader that: "Given the presence of both fixed effects and interaction terms in the difference-in-differences framework, the linear probability model estimated with ordinary least squares is more appropriate than logistic regression. For completeness, we report the odds ratio for the 1985 cohort from a conditional logit model: 3.36 (0.1-111.8). While the estimation problems with logistic in this context means we cannot interpret the magnitude, the sign is consistent with the results we report from ordinary least squares."

Ai, Chunrong and Edward C. Norton, "Interaction terms in logit and probit models," *Economics Letters* 80:123-129, 2003.

Chamberlain, Gary, "Analysis of Covariance with Qualitative Data," *Review of Economic Studies* 47:225-238, 1980.

Specific Comments (line numbers refer to actual lines of text, not the lines in the margin)

1. Page 2, line -3. I suggest shifting “adverse health and social impacts” to “adverse health and social impacts of the BGD event”.

We have made this change, thank you.

2. Page 3, Data section, first paragraph line 7. “n=1,260 births with nonmissing...”. What was the total number of births reported (so the 1,260 number can be put in context).

In fact, none of the 1981-1985 births within 250 km of Bhopal had missing data for child sex. To avoid confusion, we have removed the clause on “with nonmissing data on child sex.”

3. Page 4, line -8. “under some standard assumptions”. It would be helpful to give a general reference to difference-in-differences methods here for readers interested in more details on the standard assumptions.

We have added a citation to a standard econometrics textbook: Wooldridge, Jeffrey M., *Introductory Econometrics: A Modern Approach*, 7th edition, Cengage Learning, 2019.

4. Page 4, line -3. “below 10 years old” to “below 10 years old at the time of BGD”.

We have made the change in the text.

5. Page 5, last line of paragraph containing equation 1. “Standard errors...to account for spatial and serial autocorrelation”. I have two quick questions regarding the standard error calculations:

a. Are the standard errors also adjusted for the population size in each district? The variance of estimated disease risks, especially for rare outcomes, is dependent on the number of individuals observed.

While the referee is correct that district-level disease risk would exhibit a variance proportional to the inverse of population, in this paper we are not examining population-level risk. The adjustment you suggest would be necessary to infer a population-level effect of the BGD on the outcomes in question, but that is not what this study strives for. Since our unit of observation is the individual, our results are internally valid causal estimates without adjusting for varying population sizes across districts (Solon et al., 2013).

Solon, Gary, Steven J. Haider and Jeffrey M. Wooldridge, "What Are We Weighting For?" *Journal of Human Resources* 50(2): 301-317, 2015.

b. I believe that the clustering by birth year and district will adjust for observations with the same district-year. In a sense, this adjusts for "spatial autocorrelation" by allowing repeated-measures-type correlations between observations from the same spatial region, but (I don't believe) this is not the same as adjusting for the more typical use of the term "spatial autocorrelation" where observations are correlated to other nearby observations where nearby can mean in adjacent regions or with correlations decreasing as distance between observations increase. If this is the case, I suggest the authors adjust "for spatial and serial autocorrelation" to "for correlations between observations occurring in the same year and district".

To clarify, the technique we apply is two-way clustering, and not clustering by each birth year within district. Two-way clustering allows for robust inference when there is two-way nonnested clustering (Cameron et al., [2011]). In our context, this gives the benefit of adjusting standard errors for spatial autocorrelation within district, serial correlation within district (there is likely correlation over time in outcomes with the same district), and commonalities across people in the same birth cohort (for example due to particularly high food prices in a key year of childhood).

We have changed the sentence to make this point more clear: "Standard errors are two-way clustered by district and birth year in order to account for spatial and serial autocorrelation within district, as well as covariate shocks across all people in the same birth cohort (such as changes in food prices)."

Cameron, Colin A., Jonah B. Gelbach and Douglas L. Miller, "Robust Inference with Multiway Clustering," *Journal of Business & Economics Statistics*, 29(2): 238-249, 2011.

6. Page 6, line 2. "born in 1985" to "born in 1985 (the year following BGD)".

We have made the change.

7. Page 6, Table 1. Are the results in Table 1 limited to those living within 100 km at the time of BGD? The text suggests this is the case, but the table caption suggests the results apply to all men born in Madhya Pradesh.

These results include men surveyed throughout Madhya Pradesh, and compares men living within 100 km of Bhopal to those beyond 100 km of Bhopal. We have added table notes which now makes this clear.

8. Page 7-8. Thanks to the authors for a thoughtful discussion. Woven in the discussion of potential environmental and social drivers of risk, is an interesting point about the type of causal conclusions enabled by the causal inference approach. The results provide causal insight to the impact of the BGD *event* itself (time and location), but, as noted by the authors, the results do not provide causal insight into the impact of the MIC exposures themselves. In the expansion of the application of causal inference within epidemiology, I feel this is an important point to specifically articulate. The authors' analysis provides insight into the BGD event and the types of outcomes detail some of the broad-reaching impacts of the event on health and social impacts. The paragraph on page 8 beginning "A few study limitations..." is insightful in bringing interpretation to elements of the event. To bring this home, I suggest adding a summary statement at the end noting that the use of causal inference mitigates but does not eliminate the need to consider different routes of causation. I found the discussion here to be very helpful in bringing to light some nagging concerns regarding the need for care in interpretation of causal inference methods that I had not quite been able to articulate prior to reading the manuscript. Well done.

We thank the referee for the kind words, and have added a sentence to the discussion as suggested.

VERSION 2 – REVIEW

REVIEWER	Prakash, Nishith Northeastern University College of Social Science and Humanities, School of Public Policy and Urban Affairs
REVIEW RETURNED	24-Feb-2023

GENERAL COMMENTS	Great job at revising the draft!
----------------------------------

REVIEWER	Kumar, Santosh University of Notre Dame, Keough School of Global Affairs
REVIEW RETURNED	17-Mar-2023

GENERAL COMMENTS	Hi, I am satisfied with the revision and the authors have addressed all points.
--

REVIEWER	Waller , Lance A. Rollins School of Public Health, Biostatistics and Bioinformatics
REVIEW RETURNED	13-Mar-2023

GENERAL COMMENTS	General Comments 1. Many thanks to the authors for a careful review. Two quick points remain. a. Thank you for clarifying the linear and logistic regression point. I had been under the impression that the data represented rates from the clusters, but the authors' response suggests the data are 0 or 1 responses at the individual level. If this is the case, my apologies for misunderstanding, please make sure that this is clear in the data description. (It makes sense in hindsight, but many spatial analyses of local disease risks are based on small group counts and estimated rates (for which one could use a log transformation as long as there are no zeros or zeros are adjusted).
--

	b. In response to the note about adjustments for spatial correlation. Thank you for clarifying the adjustment for two-way clustering. While this is clearer to me know, I still feel this provides a “spatial” adjustment based on membership in the cluster within a district rather than adjusting for “spatial” correlation based on distance to neighboring observations. This distinction may be somewhat semantic, but most of the spatial autocorrelation literature either has correlation decay with distance, or defines correlation between neighboring districts, that is, correlation decreases with distance between observations. The author’s statement note that their approach adjusts for “spatial correlation within district”, but I’m not entirely convinced that this differs from adjusting for “correlation within district” (i.e., adjusting for shared cluster membership). Can this be clarified?
--	--

VERSION 2 – AUTHOR RESPONSE

Reviewer: 4

Dr. Lance A. Waller , Rollins School of Public Health

Comments to the Author:

General Comments

1. Many thanks to the authors for a careful review. Two quick points remain.
 - a. Thank you for clarifying the linear and logistic regression point. I had been under the impression that the data represented rates from the clusters, but the authors’ response suggests the data are 0 or 1 responses at the individual level. If this is the case, my apologies for misunderstanding, please make sure that this is clear in the data description. (It makes sense in hindsight, but many spatial analyses of local disease risks are based on small group counts and estimated rates (for which one could use a log transformation as long as there are no zeros or zeros are adjusted).

We are happy to make this more clear. We have added the following sentence in the Data section: “The unit of observation in all analyses is the individual.”

- b. In response to the note about adjustments for spatial correlation. Thank you for clarifying the adjustment for two-way clustering. While this is clearer to me know, I still feel this provides a “spatial” adjustment based on membership in the cluster within a district rather than adjusting for “spatial” correlation based on distance to neighboring observations. This distinction may be somewhat semantic, but most of the spatial autocorrelation literature either has correlation decay with distance, or defines correlation between neighboring districts, that is, correlation decreases with distance between observations. The author’s statement note that their approach adjusts for “spatial correlation within district”, but I’m not entirely convinced that this differs from adjusting for “correlation within district” (i.e., adjusting for shared cluster membership). Can this be clarified?

The referee is correct that clustering by district adjusts for any structure of correlation within district, including correlation that follows an explicitly spatial distance decay pattern. We have made the suggested change in order to capture the more general within-district correlation: “Standard errors are two-way clustered by district and birth year in order to account for cross-sectional and serial autocorrelation within district, as well as covariate shocks across all people in the same birth cohort (such as changes in food prices).

VERSION 3 – REVIEW

REVIEWER	Waller , Lance A. Rollins School of Public Health, Biostatistics and Bioinformatics
REVIEW RETURNED	06-Apr-2023
GENERAL COMMENTS	Thank you for the careful revision, all of my previous concerns have been addressed.